computer modelling and simulation/ecology

cellular automata modelling, computational fluid dynamics, eel tile, European eel, individual-based modelling

**Author for correspondence:**
Thomas E. Padgett
e-mail: ed10tep@leeds.ac.uk

# Individual-based model of juvenile eel movement parametrized with computational fluid dynamics-derived flow fields informs improved fish pass design

Thomas E. Padgett[1], Robert E. Thomas[3],
Duncan J. Borman[2] and David C. Mould[4]

[1]Centre for Doctoral Training in Fluid Dynamics, and [2]School of Civil Engineering, University of Leeds, Leeds, LS2 9JT, UK
[3]Energy and Environment Institute, University of Hull, Hull, HU6 7RX, UK
[4]JBA Consulting, Salts Mill, Saltaire, BD18 3LF, UK

TEP, 0000-0001-8396-2931; DJB, 0000-0002-8421-2582

European eel populations have declined markedly in recent decades, caused in part by in-stream barriers, such as weirs and pumping stations, which disrupt the upstream migration of juvenile eels, or elvers, into rivers. Eel passes, narrow sloping channels lined with substrata that enable elvers to ascend, are one way to mitigate against these barriers. Currently, studded eel tiles are a popular substrate. This study is the first to evaluate the flow fields within studded eel tiles and to model the swimming performance of elvers using cellular automata (CA) and individual- (or agent-) based models. Velocities and flow depths predicted by a computational fluid dynamics model of studded eel tiles are first validated against published values for a single installation angle–discharge combination. The validated model is then used to compute three-dimensional flow fields for eel passes at five different installation angles and three inflow discharges. CA and individual-based models are employed to assess upstream passage efficiency for a range of elver sizes. The individual-based model approximates measured passage efficiencies better than the CA model. Passage efficiency is greatest for shallow slopes, low discharges and large elvers. Results are synthesized into an easy-to-understand graphic to help practitioners improve eel pass designs.

# 1. Background

The European eel (*Anguilla anguilla*) is a catadromous species. Born in the Sargasso Sea, they are transported as larvae along the Gulf Stream, arriving on the Atlantic coast of Europe after a nine-month journey [1]. On arrival, larvae metamorphose into glass eels, or elvers, and migrate upriver [2] where they can live for up to 50 years before returning to the Sargasso Sea to spawn [3]. Recruitment of elvers to rivers across Europe has suffered a 95% reduction since the early 1980s and *Anguilla anguilla* has thus become a species of high conservation concern [4–6]. This decline has been attributed to the reduced connectivity of hydrosystems caused by a number of in-stream barriers [7,8]. In-stream barriers can be mitigated through the installation of fish passes, which facilitate the passage of aquatic species by reducing the energy of the flow. However, due to the low swimming performance of eels and their inability to jump out of water [9], common fish pass facilities such as the Larinier pass or vertical slot fishway are inefficient and inappropriate [8,10].

Anguilliform-specific passage facilities have been developed and shown to be effective in enabling the upstream passage of eels and elvers [11–13]. Eel passes comprise relatively steep ascent ramps that provide a wetted substrate designed to facilitate eel passage through crawling and swimming in near-boundary regions with lower flow velocities [14]. Historically, substrata were often cheap, robust items such as rocks, aggregates, branches [10] and burlap [15] or geotextile matting [10,16], but these were found to be too abrasive and caused passing eels to lose a considerable amount of mucus [17]. Purpose-built, synthetic substrates comprising small, more-or-less rigid, vertical cylinders or studs attached to a modular base that can be placed beside one another to create an eel pass have recently become available (e.g. [18–20]). These have been shown, in laboratory experiments of a model crump weir, to increase the passage efficiency of elvers from 0% to 67% [13]. However, those experiments were undertaken for elvers of a specific size traversing a constant, idealized, geometry and at a single installation angle [13]; extrapolation of these findings to other crump weirs, let alone different hydraulic structures, is therefore risky.

In response to the lack of empirical studies that quantitatively assess flow fields within, and the passage efficiency of, eel passes, this study aims to accomplish four things. First, the near-substrate velocity fields in a pass comprising eel tiles produced by Berry & Escott Engineering [18] are quantified using three-dimensional computational fluid dynamics (CFD) modelling. The flow structures within common fish passes have previously been studied both experimentally and through the use of CFD (see [21–24]), but the hydrodynamic structure of eel passes has not yet been quantified. Second, the performance of an example pass is assessed by repeating numerical simulations over a range of installation angles and flow rates, and comparing the resulting velocity fields against the known swimming capabilities of elvers of a range of sizes. Third, the passage efficiency of each installation angle and flow rate combination is quantified further by applying, for the first time, cellular automata (CA) and individual-based models of elver movement to fish passes. Although individual- (or agent-) based models have become popular within the field of ecology and have been applied to fish (e.g. [25,26]), these models represent the first attempt to simulate elver or eel motion in an agent-based framework. Last, results are contextualized and then synthesized into summary charts to assist practitioners to better understand the consequences of design decisions for upstream passage efficiency.

# 2. Methods

This study considers a typical eel pass constructed from multiple dual density, studded, eel tiles [18] (figure 1). These tiles feature 50 mm high studs of two different tapered diameters and centre-to-centre spacings: 14.8 mm at the base tapering to 11.7 mm at the top, with a centre-to-centre spacing of 45.45 and 29.6 mm at the base tapering to 23.4 mm at the top, with a centre-to-centre spacing of 83.3 mm. Each tile features a roughly 2 : 1 ratio of large studs to small studs (figure 1). Vowles *et al*. [13] simulated such an eel pass in a laboratory study that used a model 0.25 m high crump weir that measured 1.25 m in the streamwise ($x$) direction and 0.3 m in the cross-stream ($z$) direction. The downstream face of the weir was inclined at 11.5° from the horizontal, while the upstream face was inclined at 30° from the horizontal. The approach flow depth was 0.278 m and the nominal discharge per unit width was approximately $3.33 \times 10^{-3}$ m$^2$ s$^{-1}$ [13]. This discharge has significant uncertainty since the mean inflow velocity was reported to be $8.0 \pm 6.1$ mm s$^{-1}$ ($\mu \pm 1 \ \sigma$) [13], which yields a discharge per unit width of $2.22 \times 10^{-3} \pm 1.70 \times 10^{-3}$ m$^2$ s$^{-1}$. Since only a shallow, 5.0 mm thick, sheet

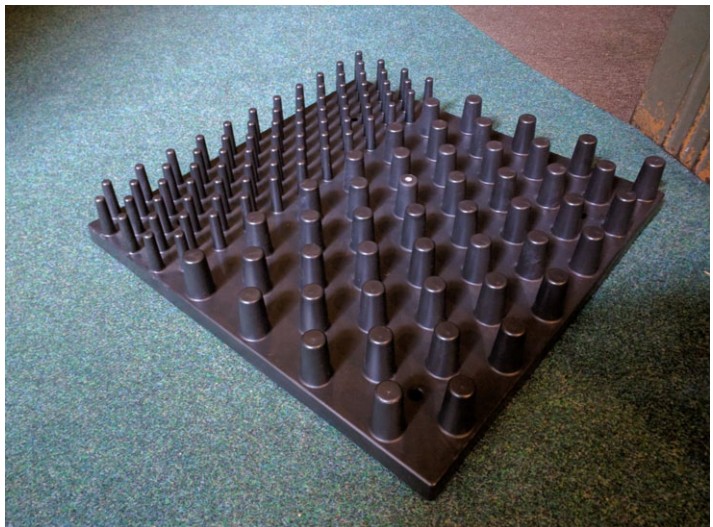

**Figure 1.** Typical dual density studded eel tile [19] (Photograph by T. E. Padgett, 14 June 2017).

of water flowed over the downstream face of the weir, the time taken for 5 ml of India ink to flow down the weir face was used to estimate the mean velocity on the weir face. Mean velocity was estimated as $0.347\,\mathrm{m\,s^{-1}}$ [13]. The mean flume water temperature was $21.8 \pm 0.96°\mathrm{C}$. Herein, the geometry employed and the values reported by Vowles *et al.* [13] are used to parametrize and assess initial CFD simulations. A parametric study is then undertaken using installation angles of 8°, 11°, 14°, 17° and 20° and inflow discharges per unit width of $1.67 \times 10^{-3}$, $3.33 \times 10^{-3}$ and $5.0 \times 10^{-3}\,\mathrm{m^2\,s^{-1}}$ to assess the ability of elvers to ascend studded eel passes.

## 2.1. Computational fluid dynamics methodology

Since the smaller of the two stud sizes was favoured by ascending elvers [13], ANSYS Fluent v. 17.2 [27] was used to construct a CFD model based only on the geometry of the smaller studs (figure 2). To facilitate this, it was assumed that the discharge was distributed across the tile according to the ratio of the centre-to-centre stud spacings, 45:83, and so a larger proportion of the discharge was assumed to pass through the larger studs. Only simulating the smaller studs simplified the computational mesh, which was simplified further by introducing symmetry boundary conditions at one-third and two-thirds of the channel width in the spanwise ($z$) direction, reducing the size of the computational domain by two-thirds. The geometry required the use of an unstructured tetrahedral mesh featuring approximately 906 000 cells, with increased cell density at the studs and at the bed (figure 2b). To adequately capture the boundary layer, an 'inflation layer' [27] was applied to the bed and to the walls of each stud. This split the near-wall region into five layers of cells, each of which was 1.2 times larger than the previous cell layer. Thus, cell layer thickness transitioned smoothly from a near-wall value of approximately 0.4 mm to a value of 1.0 mm at a distance of 3.0 mm from the wall. Simulations were performed using the unsteady, incompressible, Reynolds-averaged Navier–Stokes (RANS) equations:

$$\frac{\partial \overline{u_i}}{\partial x_i} = 0 \tag{2.1}$$

and

$$\frac{\partial \overline{u_i}}{\partial t} + \frac{\partial}{\partial x_j}(\overline{u_i u_j}) - g_i + \frac{1}{\rho}\frac{\partial P}{\partial x_i} - \nu \frac{\partial^2 \overline{u_i}}{\partial x_j^2} + \frac{\partial}{\partial x_j}(\overline{u_i' u_j'}) = 0, \tag{2.2}$$

where equation (2.1) is the mass conservation equation and equation (2.2) is the momentum conservation equation, and $u$ = velocity $(\mathrm{m\,s^{-1}})$, $x$ = displacement (m), $t$ = time (s), $\rho$ = density of water (approx. $997.8\,\mathrm{kg\,m^{-3}}$ at 21.8°C), $g$ = gravitational acceleration vector $[-g\sin\theta, -g\cos\theta, 0]$, $P$ = pressure (Pa), $\nu$ = kinematic viscosity of water (approx. $9.6 \times 10^{-7}\,\mathrm{m^2\,s}$ at 21.8°C), the indices $i$ and $j$ ($i, j$ = 1 to 3) denote the three components of displacement ($x$ = slope-parallel, $y$ = slope-perpendicular, $z$ = spanwise) and velocity ($u$, $v$, $w$), overbars denote time averages and primes denote fluctuations about those

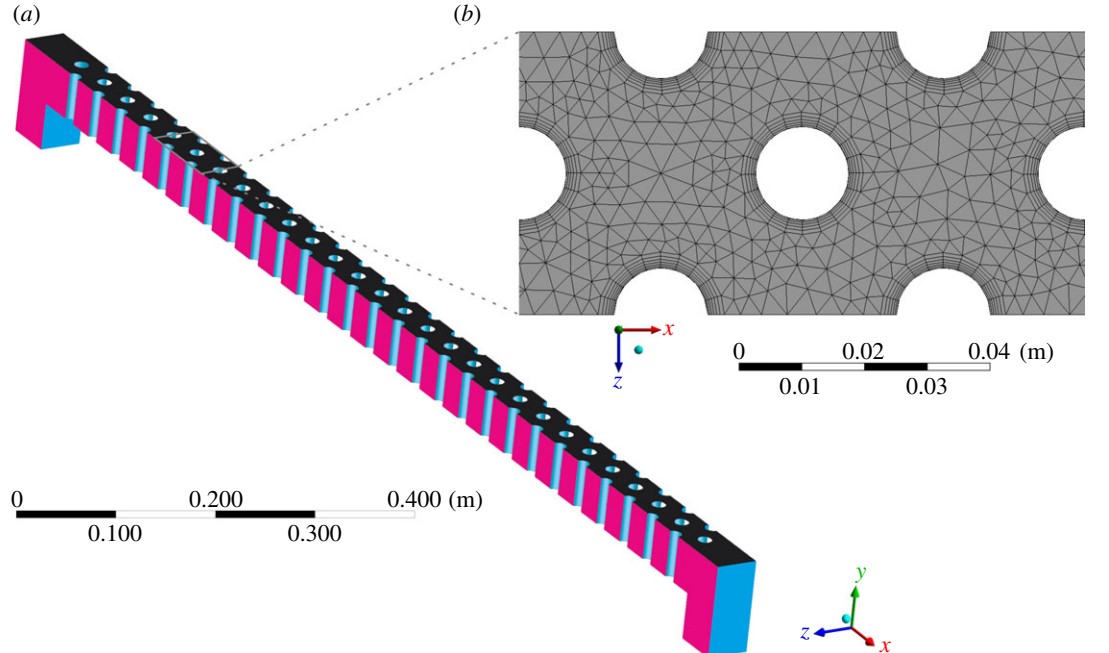

**Figure 2.** (*a*) Isometric view of the CFD domain. Cyan/light grey denotes no-slip boundary condition, magenta/dark grey denotes symmetry boundary condition and black denotes pressure outlet boundary condition and (*b*) magnified planform view highlighting the unstructured tetrahedral mesh and grid refinement near studs and walls.

averages. The momentum conservation equations require a turbulence closure for the Reynolds stress term $\partial/\partial x_j(-\rho\overline{u_i'u_j'})$. The $k$–$\omega$ *SST* model was used [28], which combines the robust and accurate formulation of the $k$–$\omega$ model in the near-wall region with the free-stream independence of the $k$–$\varepsilon$ model in the far field [29]. The $k$–$\omega$ shear stress transport (*SST*) model exhibits enhanced performance relative to conventional $k$–$\varepsilon$ and $k$–$\omega$ models when applied to adverse pressure gradient flows, aerofoils and sub- to super-critical transitions [29]. The temporal gradients and the advection terms in equations (2.2), the turbulent kinetic energy, $k$, and specific turbulence dissipation rate, $\omega$, were discretized using a second-order upwind scheme. The SIMPLE scheme [30] was used to couple the velocities and the pressure. The free surface location was approximated using the volume of fluid (VOF) method [31] with two Eulerian phases, water and air, and using the implicit body force formulation. The free surface was interpolated using Fluent's Geo-Reconstruct scheme, which fits a piecewise-linear interface within each cell and uses that linear shape to estimate the advection of fluid through the cell faces in a three-step procedure [27].

Water entered the domain through a velocity inlet positioned upstream of the eel pass and left the domain through a pressure outlet located downstream of the pass (figure 2). All other boundaries were defined as walls with a no-slip boundary condition and a roughness of zero. A gravitational acceleration term, with components in the $x$ and $y$ directions to define the installation angle, was applied to the domain. Incorporating the gravitational acceleration term in this manner permitted the ready solution of multiple installation angles without the need to remesh. Simulations were undertaken with a constant timestep, $\Delta t$, of $5.0 \times 10^{-4}$ s. The convergence criterion for the non-dimensional residuals of $u$, $v$ and $w$, mass continuity, turbulent kinetic energy, $k$, and specific turbulence dissipation rate, $\omega$, was defined as $10^{-4}$. A maximum limit of 100 iterations per timestep was imposed.

## 2.2. Quantifying eel tile 'passability'

In order to assess the 'passability' of eel tiles at different flow rates and inclinations, it was necessary to compare computed three-dimensional flow fields against the swimming and/or climbing abilities of elvers. In the absence of data detailing climbing performance, the burst swimming performance data of Clough *et al.* [32] were used. These data were obtained from 417 elvers collected from the River Severn in April (mean length $67.61 \pm 0.57$ mm; mean water temperature $11.1 \pm 0.32°C$) and in June/ July (mean length $147.76 \pm 4.81$ mm; mean water temperature $18.64 \pm 0.12°C$) [32]. Clough *et al.* [32] summarized their results in the SWIMIT 3.3 model, which extracts burst swimming speeds of up to

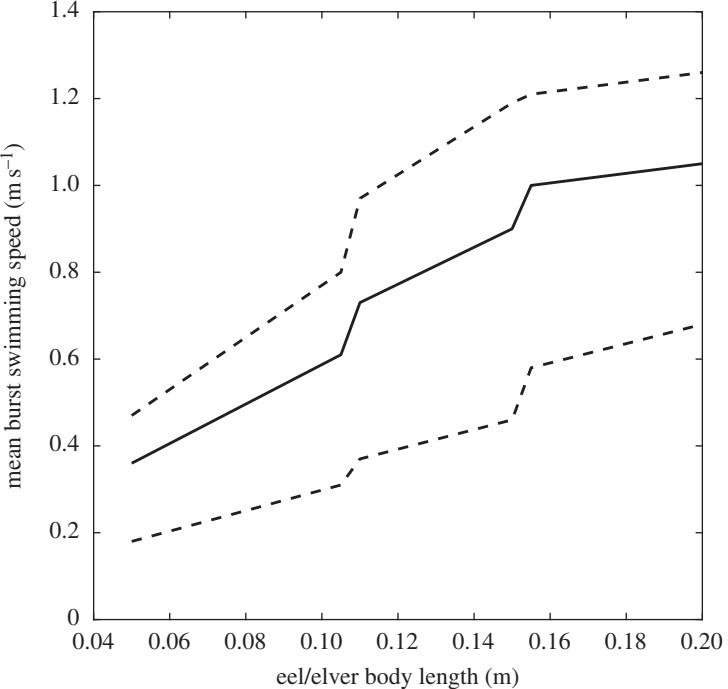

**Figure 3.** Median, 10th and 90th percentile burst swimming speed in spring, at a water temperature of 21.8°C against eel and elver body length from the SWIMIT 3.3 model [32]. The solid line depicts the median burst swimming speed while dotted lines show the 10th and 90th percentile burst swimming speeds.

0.08 m long elvers from lookup tables and estimates median, 10th and 90th percentile burst swimming speeds of larger fish using equations derived from multiple regression [32]. The availability of median (i.e. average) swimming speeds necessitated temporal averaging of the CFD-derived flow fields. Preliminary simulations indicated that a total flow duration of 8 s was sufficient to permit incoming fluid to exit the domain; a simulation duration of 10 s was selected to permit time-averaging of at least 2 s of data after complete inundation of the domain. In addition, those simulations indicated that maximum flow velocities occurred 3 mm above the floor; this height also coincided well with the estimated mean diameter of the elvers tested by Vowles *et al.* [13]. Thus, for each case, velocity fields were extracted from a plane parallel to and 3 mm above the tile bed and temporally averaged over the 2 s period between 8 and 10 s of flow time. The domain was then mirrored in the spanwise direction, returning the domain to its original width (0.135 m). The resulting velocity fields were linearly interpolated from their irregular grids onto a regular $0.5 \times 0.5$ mm grid and then classified into three classes: 'passable', 'impassable' and 'boundary' using the median burst swimming speeds [32] (figure 3). A passable grid cell was defined as one wherein the time-averaged velocity was less than the median burst swimming speed of the length of elver being considered. An impassable grid cell was defined as a cell wherein the velocity was greater than or equal to this threshold. A boundary grid cell was defined as a cell without a velocity (i.e. containing no water) that an elver could never physically pass through, such as the studs or wall. This classification process was undertaken for five installation angles (8°, 11°, 14°, 17° and 20°), three flow discharges per unit width ($1.67 \times 10^{-3}$, $3.33 \times 10^{-3}$ and $5.0 \times 10^{-3}$ m$^2$ s$^{-1}$) and the median, 10th percentile and 90th percentile burst swimming speeds for elvers of six different lengths (0.05, 0.06, 0.07, 0.08, 0.09 and 0.10 m), resulting in 270 classified domains (e.g. figure 4). This range of elver lengths encompasses the largest and smallest lengths of immigrating elvers recorded at an example European coastal region [33]. The resulting classified maps provide a useful visual indicator of the likely 'passability' of the eel pass at specific inclinations and discharges, but do not provide a quantitative metric for comparison among cases nor do they provide a metric that is readily used by practitioners.

## 2.3. Cellular automata and individual-based modelling

In order to provide passage efficiency values of elvers, and thus enable comparison with the results of Vowles *et al.* [13] and wider literature, CA and individual-based models were developed.

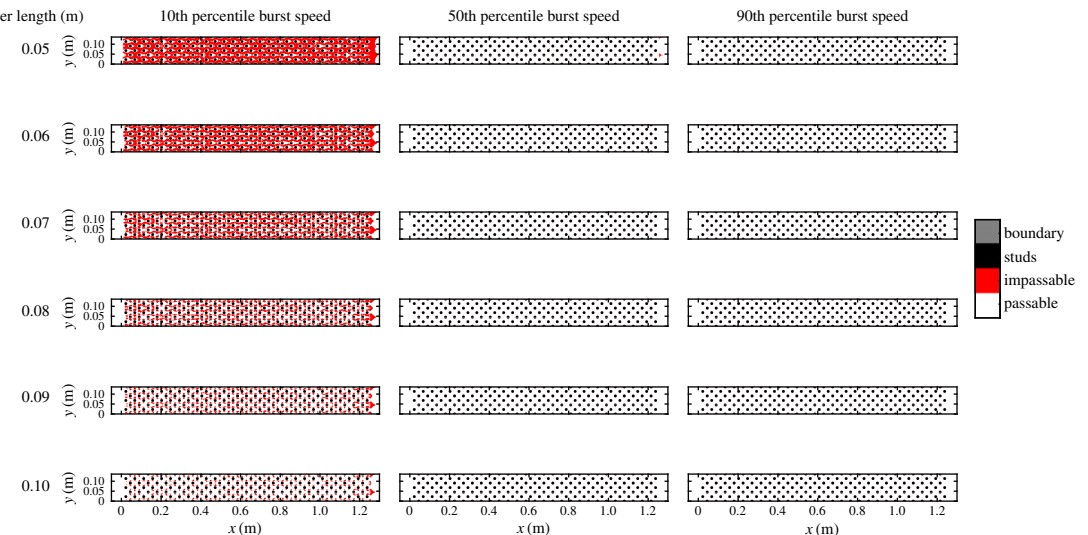

**Figure 4.** Classified time-averaged velocity fields extracted in a plane 3 mm above the floor of an eel tile installed at 11° at a discharge per unit width of $3.33 \times 10^{-3}$ m$^2$ s$^{-1}$. Classified using the 10th, 50th and 90th percentile burst swimming speeds for elver of length 0.05, 0.06, 0.07, 0.08, 0.09 and 0.10 m. Elver length (in m) is shown to the left. White denotes 'passable', red denotes 'impassable', black denotes 'studs' and grey denotes 'boundary'. Note that boundaries are one cell thick and so appear as lines in the figure, and studs are coloured separately for clarity.

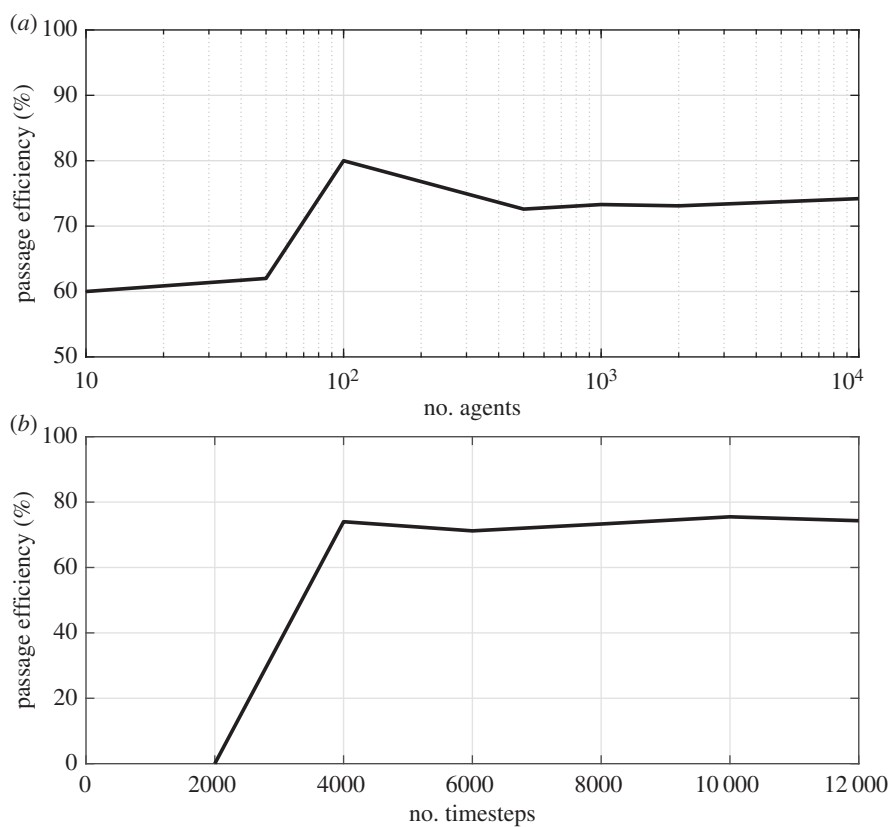

**Figure 5.** Sensitivity of passage efficiency of cellular automata (CA) agents to: (a) number of spawned CA agents and (b) number of timesteps for elver of length 0.07 m ascending an eel pass installed on a 1.25 m long model crump weir inclined at 11° and with a unit discharge of $3.33 \times 10^{-3}$ m$^2$ s$^{-1}$. All simulations in (a) were performed over 10 000 timesteps, while all simulations in (b) were performed using 1000 CA agents.

The CA model first imported a domain with cells that had been classified as either 'passable', 'impassable' or 'boundary' (e.g. figure 4). Second, following sensitivity analysis (figure 5a), 1000 automata that each occupied a single cell were randomly spawned at the centroids of cells

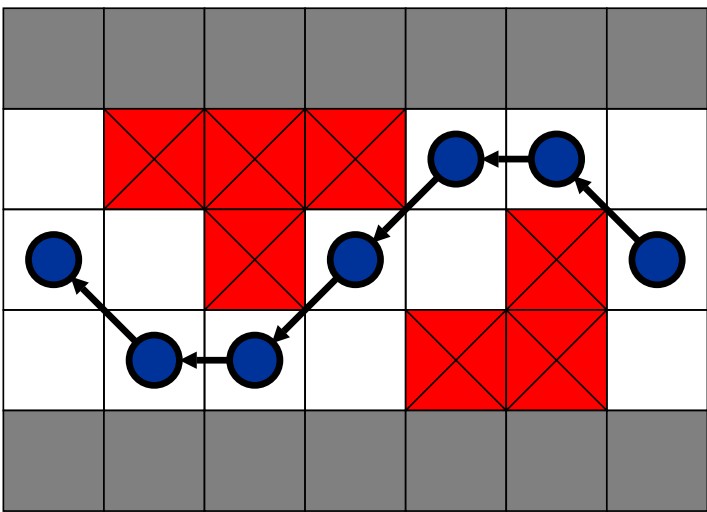

**Figure 6.** Example automaton movement. Blue circles represent automata, white squares denote 'passable' cells, red squares with black crosses denote 'impassable' cells and grey squares denote 'boundary' cells.

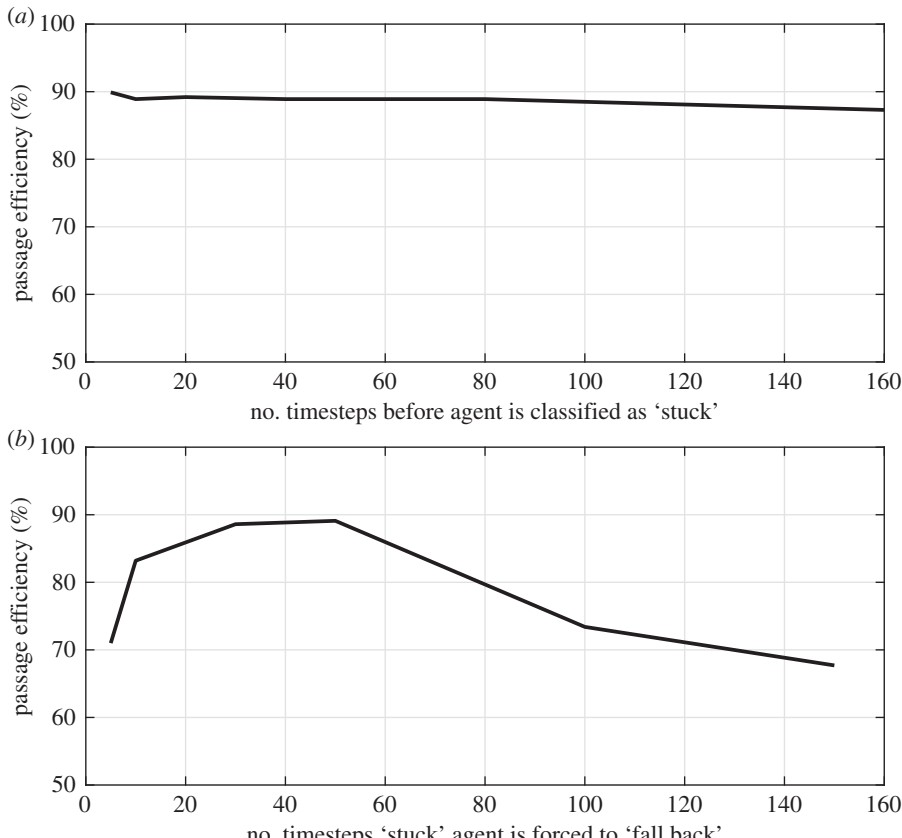

**Figure 7.** The sensitivity of passage efficiency of cellular automata (CA) agents to: (*a*) number of timesteps before a CA agent is classified as 'stuck' and (*b*) number of timesteps a CA agent that is classified as 'stuck' before it is forced to 'fall back' before recommencing ascent. Both simulations performed for elver of length 0.07 m ascending an eel pass installed on a 1.25 m long model crump weir inclined at 17° and with a unit discharge of $5.0 \times 10^{-3}$ m² s⁻¹, using 1000 agents and 10 000 timesteps; simulations in B performed with the number of timesteps before an agent is classified as 'stuck' set equal to 20.

forming the downstream-most edge of the domain, although following the sequential nature of elver emplacement within the Vowles *et al*. [13] experiments, automata did not encounter or interact with each other. Third, a first-order Moore neighbourhood was established for each automaton in order to define which cells were its neighbours [34]. A list of passable neighbours was then compiled, from

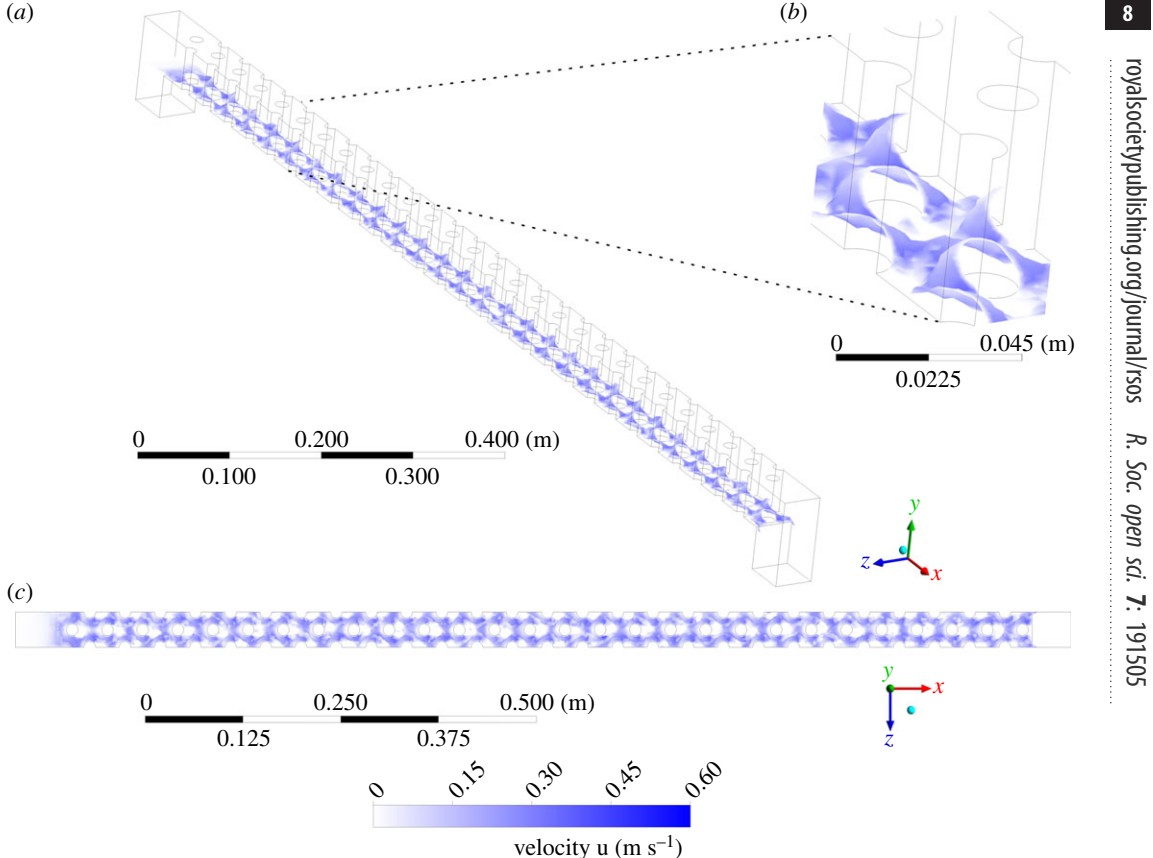

**Figure 8.** (a) Isometric, (b) magnified isometric and (c) plan views of the pass with an installation angle of 11° and a discharge per unit width of $3.33 \times 10^{-3}$ m² s⁻¹, with overlain free surface after 10.0 s of flow time. The free surface is overlain with contours of velocity magnitude at the free surface; the velocity contour scale is consistent across the three views. The domain has been cropped to increase ease of viewing.

which a destination neighbour was randomly selected, prioritizing upstream neighbours first, cross-stream neighbours second and downstream neighbours third. Stochasticity was handled using the *random* python package, which is based on the Mersenne Twister pseudo-random number generator and has a periodicity of $2^{19937} - 1$ [35]. Fourth, automata moved to the centroid of the selected destination cell. Steps 3 and 4 were repeated sequentially for each automaton during each timestep, until all automata passed or until a maximum number of timesteps (10 000 was selected for the simulations shown herein; figure 5b for sensitivity analysis) had been reached. It is possible for an automaton to enter a cul-de-sac, from which it can migrate no further (figure 6). Therefore, following sensitivity analysis (figure 7a), a rule was added to assess the streamwise distance covered by each automaton during 20 moves and if that distance was ≤2 cells, the automaton was classified as 'stuck'. The movement priorities of a 'stuck' automaton switched to prioritize moving downstream, then cross-stream, then upstream and thus an automaton could 'fall back'. Sensitivity analysis indicated that maximum passage efficiency occurred when automata were forced to make 30–50 fall-back moves (figure 7b), and so to minimize run times, a value of 30 was selected for this parameter. It was, therefore, possible for automata to make multiple attempts to pass obstacles.

Although fall-back behaviour has been observed in eels and elvers [13,36], the CA model did not adequately account for exhaustion: unless the maximum number of cell-to-cell moves was exceeded, an elver continued to make passage attempts ad infinitum. Therefore, an individual-based approach was developed to explicitly account for exhaustion. This model adopted the same overall structure as the CA model, but with some significant differences. First, the unclassified time-averaged velocity field at 3 mm above the floor for the selected eel pass configuration (e.g. figure 8c) was imported rather than the classified 'passability' domain. Second, each individual agent was randomly assigned a burst swimming speed using the median, 10th and 90th percentiles and the inverse lognormal distribution that fitted those data for each length of elver [32] (figure 3). The movement

mechanism and selection of neighbours was the same as for the CA model, except that during each movement, the relative velocity of each agent was calculated by differencing its burst swimming speed and the water velocity at the centroid of the destination cell. The time required for an elver to make each move was then calculated and collated, and an elver was classified as exhausted if the total time exceeded 20 s, which is the standard time period over which it is possible for fish to sustain burst swimming speeds [32].

For both the CA model and the individual-based model, an automaton or agent was classified as having successfully passed the domain once it reached the upstream-most edge of the domain. If an automaton or agent was still within the domain either once it became exhausted or the maximum number of timesteps was exceeded, it was classified as a failed passage.

# 3. Results

## 3.1. Three-dimensional flow fields

Simulated flow fields are shown in figures 8a–c. Mean velocity and flow depth for the 11°, $3.33 \times 10^{-3} \, \mathrm{m^2 \, s^{-1}}$ case compare favourably against those reported for a similar case by Vowles *et al.* [13]: $0.299 \, \mathrm{m \, s^{-1}}$ versus $0.347 \, \mathrm{m \, s^{-1}}$ and 5.58 mm versus 5.0 mm. These simulated values yield Reynolds numbers of 1740 or 4550 using the mean flow depth or the stud diameter as the length scale, respectively. In other words, the flow is fully turbulent [37,38]. Indeed, flows are fully turbulent for all simulated cases, with Reynolds numbers ranging from 660 or 2320 for the 8°, $1.67 \times 10^{-3} \, \mathrm{m^2 \, s^{-1}}$ case to 3070 or 5800 for the 20°, $5.0 \times 10^{-3} \, \mathrm{m^2 \, s^{-1}}$ case using the mean flow depth or stud diameter as the length scale, respectively. In planform, flow patterns are thus as expected for a turbulent fluid flowing through a field of vertical cylinders: water decelerates as it approaches each stud, with a stagnation zone at the upstream face of each stud, followed by acceleration where flow converges between the studs and strong shedding of wakes from the downstream face of each stud (figures 8c,9a). However, the spacing between studs is such that there is a strong interaction between wakes and proceeding studs (e.g. [37,38]), so there is insufficient time or space for a Kármán vortex street to develop. In the cross-stream direction, streamwise- and time-averaged velocities increase from minima of $0.26 \, \mathrm{m \, s^{-1}}$ in regions shielded by studs, corresponding to the no-slip boundary condition, to maxima of approximately $0.37 \, \mathrm{m \, s^{-1}}$ at one-quarter and three-quarters of the width of the channel (figure 9a), corresponding to regions between studs. Conversely, streamwise-averaged water depth decreases from maxima of approximately 6.5 mm in regions shielded by studs to minima of approximately 4.6 mm at one-quarter and three-quarters of the width of the channel (figures 8b and 9b).

## 3.2. Classified flow fields

Example classified flow fields for the 11°, $3.33 \times 10^{-3} \, \mathrm{m^2 \, s^{-1}}$ case are shown in figure 4. Additional classified flow fields can be found in the electronic supplementary material. It is clear that the difference between the 10th and 50th percentile burst swimming speeds is significant and sufficient to cause much of the pass to be impassable by elvers capable of the 10th percentile burst swimming speeds; only the narrow region surrounding the studs is passable by elvers less than 0.08 m long. The swimming ability of elvers improves as their length increases and thus less of the pass is impassable and more of the pass is passable as elver length increases (figures 3 and 4). Assuming the 10th percentile burst swimming speeds, only 34% of the area of the pass is passable for 0.05 m long elvers, while 90% of the pass is passable for 0.10 m long eels. However, employing the percentage of the domain that is passable in this manner does not capture the likelihood of an elver ascending the pass. For example, 58.5% of the domain is passable by 0.07 m long elvers but, since there is no continuous passable path from the downstream end of the pass to the upstream end of the pass (figure 4), it is not possible for a 0.07 m long elver to ascend. Elvers longer than 0.08 m should be able to ascend the pass, assuming that they are capable of the 10th percentile burst swimming speed. Using the 50th or 90th percentile burst swimming speeds, it is possible for elvers of all tested lengths to ascend the pass at an installation angle of 11° and a discharge per unit width of $3.33 \times 10^{-3} \, \mathrm{m^2 \, s^{-1}}$ (figure 4).

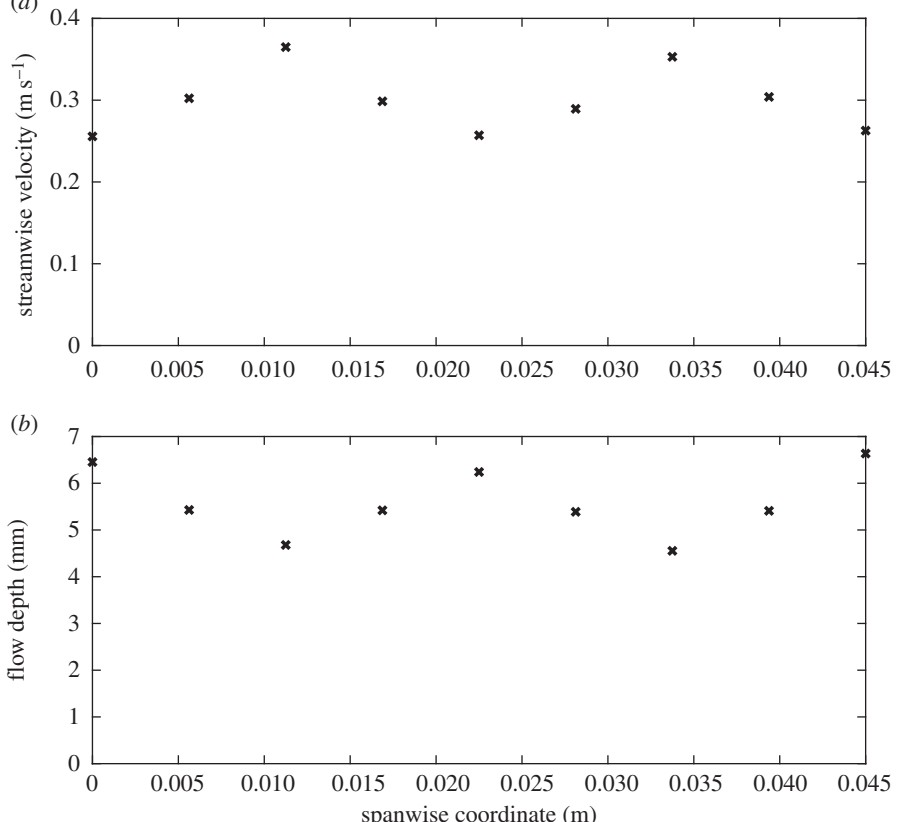

**Figure 9.** (*a*) Streamwise-averaged streamwise velocity at a height of 3 mm above the bed and (*b*) streamwise-averaged flow depth against spanwise location for the small studs of the eel tiles used by Vowles *et al*. [13]. Data have been temporally averaged between 8 and 10 s of flow time.

## 3.3. Model results

### 3.3.1. Cellular automata model

The passage success rate of automata increases with elver size and decreases with increasing installation angle (figure 10). Assuming that elvers are capable of the 10th percentile burst swimming speeds, only elvers longer than 0.08 m are able to ascend the pass, irrespective of the discharge per unit width. Assuming that elvers are capable of the 50th percentile burst swimming speeds, in addition to the noted trends of passage success with elver size and installation angle, passage success also generally decreases with increasing discharge. Any deviation from these trends is thought to be due to numerical instabilities within the CFD solutions caused by the combination of very small flow depths and fast velocities at a discharge per unit width of $1.67 \times 10^{-3} \, \mathrm{m^2 \, s^{-1}}$ (e.g. the passage success rate at an installation angle of 17°; figure 10). At a discharge per unit width of $3.33 \times 10^{-3} \, \mathrm{m^2 \, s^{-1}}$ and installation angle of 20°, 0.05 m long elvers have a passage success rate of 0%, while 0.06 m long elvers have a passage success rate of 47.8% (figure 10). At a discharge per unit width of $5.0 \times 10^{-3} \, \mathrm{m^2 \, s^{-1}}$ and installation angle of 20°, 0.05 and 0.06 m long elvers have passage success rates of 0%, while 0.07 m long elvers have a passage success rate of 86.2% (figure 10). The success rate of 0.06 m long elvers increases to 68.1% at an installation angle of 17° and to 99.6% at an installation angle of 14°, while the success rate of 0.05 m long elvers is 0% at an installation angle of 17° but increases to 27.8% at an installation angle of 14° (figure 10). Assuming that elvers are capable of the 90th percentile burst swimming speeds, passage success rates are all greater than 90% (figure 10).

### 3.3.2. Individual-based model

The individual-based model yields results that are much less binary in nature than the CA model (figure 10), reflecting the natural variability of burst swimming abilities within a population of elvers. At a discharge per unit width of $1.67 \times 10^{-3} \, \mathrm{m^2 \, s^{-1}}$ and an installation angle of 8°, passage success

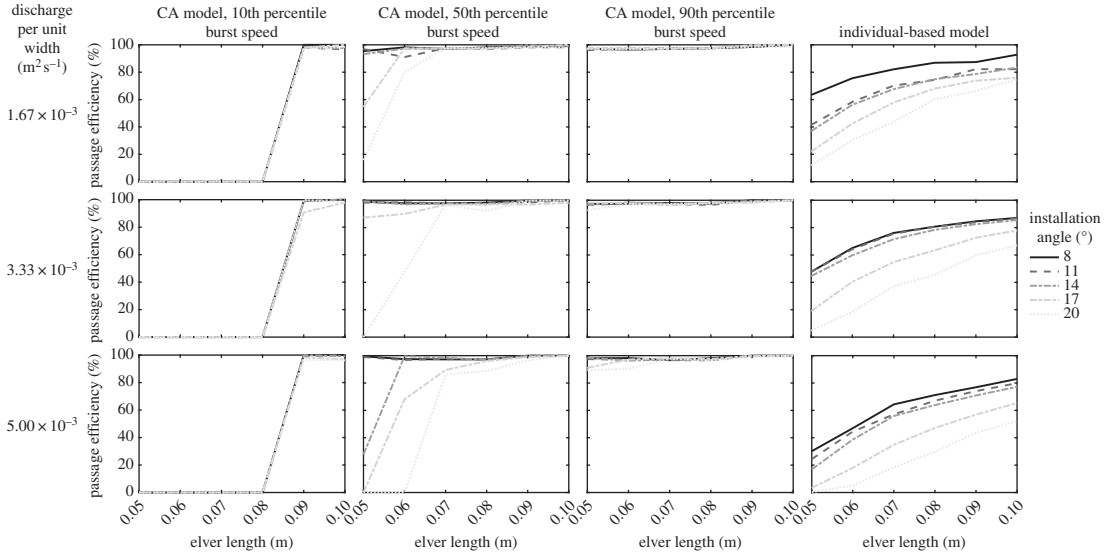

**Figure 10.** Passage efficiency (in %) predicted by the CA model assuming 10th, 50th and 90th percentile burst swimming speeds and the individual-based model against elver length for eel tiles installed at a range of installation angles at three different discharges per unit width.

**Table 1.** Passage efficiency estimated using the cellular automata and individual-based models of elvers of length 0.07 m ascending a 1.25 m long eel pass inclined at 11° and with an inflow discharge per unit width of $3.33 \times 10^{-3}$ m² s⁻¹, compared to the passage efficiencies reported by Vowles et al. [13].

| | | percentile eel burst swimming speed | | |
|---|---|---|---|---|
| model | | 10% | 50% | 90% |
| cellular automata | | 0.0% | 97.8% | 97.8% |
| individual-based | | | 75.5% | |
| Vowles et al. [13] | total pass | | 66.7% | |
| | small studs only | | 73.8% | |

rates vary from 63.4% for 0.05 m long elvers to 92.8% for 0.10 m long elvers (figure 10). At an installation angle of 14°, success rates reduce to vary from 37% for 0.05 m long elvers to 83.4% for 0.10 m long elvers (figure 10) and at an installation angle of 20°, success rates reduce further to vary from 11.8% for 0.05 m long elvers to 74.8% for 0.10 m long elvers (figure 10). At larger discharges per unit width, passage success rates are lower than at smaller discharges per unit width. For example, at a discharge per unit width of $5.0 \times 10^{-3}$ m² s⁻¹ and an installation angle of 8°, passage success rates vary from 30.2% for 0.05 m long elvers to 83.0% for 0.10 m long elvers (figure 10). Similarly, success rates vary from 16.9% for 0.05 m long elvers to 77.3% for 0.10 m long elvers at an installation angle of 14° (figure 10), while they vary from 0.2% for 0.05 m long elvers to 52.2% for 0.10 m long elvers at an installation angle of 20° (figure 10).

### 3.3.3. Comparison of models with Vowles et al. [13]

Within their experiments, Vowles et al. [13] performed 10, 10 min long, trials where 30 elvers (length 71.73 ± 3.87 mm) were initially placed downstream of an anguilliform pass installed at an angle of 11.5° within a flume with a discharge per unit width of $3.33 \times 10^{-3}$ m² s⁻¹ (water temperature: 21.8 ± 0.96°C). Passage efficiency, defined as the number of successes divided by the number of attempts, where an attempt commenced when the head of an elver progressed onto the downstream-most tile, was 67% [13]. However, they also reported that elvers found more success in ascending the small studs (14.1 ± 4.86 attempts; 11.7 ± 2.9 successes) compared to the large studs (12.3 ± 3.47 attempts; 8.3 ± 2.6 successes). This results in individual passage efficiencies for the large and small studs of 67.5% and 83.0%, respectively. Furthermore, 3.5 ± 1.65 attempts were assigned to the 'centre' of the

pass, with no corresponding successes [13]. If these attempts are evenly distributed between the large and small studs, the individual passage efficiencies for the large and small studs become 59.1% and 73.8%, respectively. This compares to passage success rates of 0.0%, 97.8% and 97.8% predicted by the CA model assuming 0.07 m long elvers are capable of the 10th, 50th and 90th burst swimming speeds, respectively (table 1). In comparison, the individual-based model predicts a passage success rate of 75.5% for this case (table 1), highlighting the effectiveness and promise of the individual-based modelling approach.

## 4. Discussion

This paper aimed to:

(1) quantify the near-substrate velocity fields in an example eel pass using CFD modelling;
(2) assess the effectiveness of that pass by repeating numerical simulations over a range of installation angles and flow rates, and comparing the resulting velocity fields against the known swimming capabilities of elvers of a range of sizes; and
(3) quantify the passage efficiency of each installation angle and flow rate combination by applying, for the first time, CA and individual-based models of elver movement to fish passes.

In this discussion, we address these aims and critically assess how our findings compare and contrast with existing literature, concluding by providing summary diagrams for practitioners to assist with the design process of eel passes.

First, CFD shows promise in being able to simulate the flow patterns through eel tiles; mean velocities and flow depths matched closely those observed in the experiments of Vowles *et al.* [13]. No previous studies have quantified the detailed flow fields within eel tiles and thus it is challenging to validate small-scale flow features. However, previous numerical modelling studies of the flow through analogous tube bundles suggest that two-equation turbulence closure models, such as the $k$–$\varepsilon$ model [39], the $k$–$\omega$ model [40] and the $k$–$\omega$ SST model employed herein, under-predict turbulence quantities and recirculation lengths [37,41]. Nevertheless, the $k$–$\varepsilon$ model, which the $k$–$\omega$ SST model tends to away from walls, returns reasonable predictions of the mean velocities [41] which are input into the CA and individual-based models. Note that if employed in isolation, the $k$–$\varepsilon$ model severely under-predicts the flow within separation zones [42], while the $k$–$\omega$ model overcomes this problem but suffers from over-sensitivity to the free-stream boundary condition. As alternatives to classical two-equation models, Reynolds stress models, large eddy simulation and direct numerical simulation may produce more accurate flow fields (e.g. see [37,41]), but they introduce increased complexity, associated convergence difficulties and significant computational cost [42]. If eels and elvers react to and/or interact with larger-scale flow structures (e.g. [12,43]), this level of complexity is not warranted. Furthermore, the significance of any small-scale turbulent structures produced by these more complex turbulence models would be lessened due to the temporal averaging that was necessary to create input datasets for the CA and individual-based models.

Second, it is clear that the effectiveness of an eel tile in enhancing passage efficiency is a function of a range of factors that influence the ability of eels or elvers to ascend a pass. The extensive review of Solomon & Beach [9] highlighted many factors controlling passage efficiency, including engineering parameters such as inflow discharge per unit width, flow depth, installation angle and pass length and biological parameters such as water temperature, season and fish age (and hence size, swimming and climbing ability). It is curious that this guidance was withdrawn on 12 May 2016 and is not cited heavily in the updated UK Environment Agency eel pass manual [16], which does not mention these crucial factors. The present study has investigated the influence of installation angle, discharge per unit width and elver or eel life stage; in agreement with past research [9,17,32], simulations suggest that passage efficiency increases as elver or eel length increases and as installation angle and discharge per unit width decrease.

Third, the individual-based model employed herein compares well to the results of Vowles *et al.* [13] and thus it has utility in converting maps of passable and impassable regions of a pass into passage efficiency statistics. A comparison of the resulting passage efficiencies of all configurations of eel tile show that this model, which captures the natural variation in the swimming performance of elvers, gives consistently less extreme results and a much smaller range of values when compared to the

simpler CA model. This is reflective of the fact that the individual-based model uses heterogeneous agents compared to the homogeneous automata used in the CA model. Homogeneous automata polarize results, since it is likely that if one automaton can pass, all automata can pass and vice versa, particularly if given the ability to make multiple attempts.

In comparing the CA model to the individual-based model, it is seen that passability decreases when exhaustion is modelled. This suggests that elvers become exhausted even with relatively shallow installation angles and a relatively short length of pass (1.25 m). This finding agrees with the statement of Solomon & Beach [9, p. 12] that '…we must take account not just of maximum swimming speed, but also of the ability to maintain certain swimming speeds for long enough to ascend a pass that may be many metres in length'. While the exhaustion metric employed within the individual-based model is simplistic and neglects the energy expended by eels and elvers in overcoming the drag they experience, it has highlighted the significance of passage duration in controlling passage efficiency. Specifically, passage duration is a function of pass length and relative velocity, itself a function of fluid velocity and burst swimming speed. Fluid velocity is a function of inflow discharge, pass roughness and pass installation angle.

## 4.1. Model limitations

A number of factors may have influenced model performance. First, although Vowles *et al.* [13] reported discharge per unit width as $3.33 \times 10^{-3}$ m$^2$ s$^{-1}$, our calculations estimate that discharge per unit width was $2.22 \times 10^{-3} \pm 1.70 \times 10^{-3}$ m$^2$ s$^{-1}$. Second, the experimental data of Vowles *et al.* [13] was determined for marginally larger elvers (although within 0.5 standard deviations of the mean). Third, both the CA and individual-based models assume that elvers only occupy a single cell measuring $0.5 \times 0.5$ mm. This potentially allows automata and agents to move through one-cell-wide passages that live elvers would not be able to traverse. Fourth, owing to the nature of the burst swimming speed data presented by Clough *et al.* [32], the flow fields within the eel pass were temporally averaged and thus neither account for flow unsteadiness nor the turbulence associated with it or the studs within the eel tiles. Time-averaging simplified the models and removed the possibility of time-dependent passage efficiencies. In the present context, flow unsteadiness means that the computed flow fields vary through time. As water passes through a field of vertical cylinders, a pair of counter-rotating vortices are shed at a regular frequency from each stud and advected downstream (e.g. [37,38,41]). Incorporating unsteadiness within the CA and individual-based model frameworks would thus introduce significant temporal sensitivity to the passage initiation time and the time taken (or route taken) to reach each individual cell and how that compares to the vortex shedding cycle (i.e. whether the flow is locally accelerated or decelerated by a vortex). To achieve this, the rules that define automaton and agent behaviour would require significant modification to enable an automaton or agent to hold station until the local velocity was sufficiently slow. Incorporating unsteadiness and turbulence should increase the tortuosity of passage routes and increase the time to ascend. Fifth, while it is possible that elvers would exhibit crawling behaviour or rest within the pass, neither of which are captured by the models, Vowles *et al.* [13,36] did not observe either of these behaviours in their experiments, and instead elvers attempted to ascend the eel tiles as quickly as possible using anguilliform swimming. Conversely, elvers and small eels less than 0.1 m long have been observed to climb sloping or even vertical wetted surfaces, especially if they are covered in moss or algae, at temperatures greater than 12–14.5°C [9,17,44–46]. Furthermore, the model does not account for many other factors important in successful passage such as predation within the pass, the ability for elvers to locate the pass in a timely manner, or whether the elvers are motivated to ascend the pass. Nevertheless, despite these limitations, the individual-based model results are both encouraging and promising; a relatively simple individual-based model provides reasonable predictions of the passage success of elvers through an eel pass.

## 4.2. Implications for eel pass design

The installation angle of an eel pass commonly reflects a trade-off between restricting water velocities to a comfortable range for ascending elvers (i.e. shallow installation angle) and, especially at in-stream barriers with large hydraulic heads, restricting the overall length of the pass (i.e. steep installation angle) [9]. However, to date, installation guidance largely originates from the observations of practitioners and the recommendations of manufacturers; there is little objective empirical or

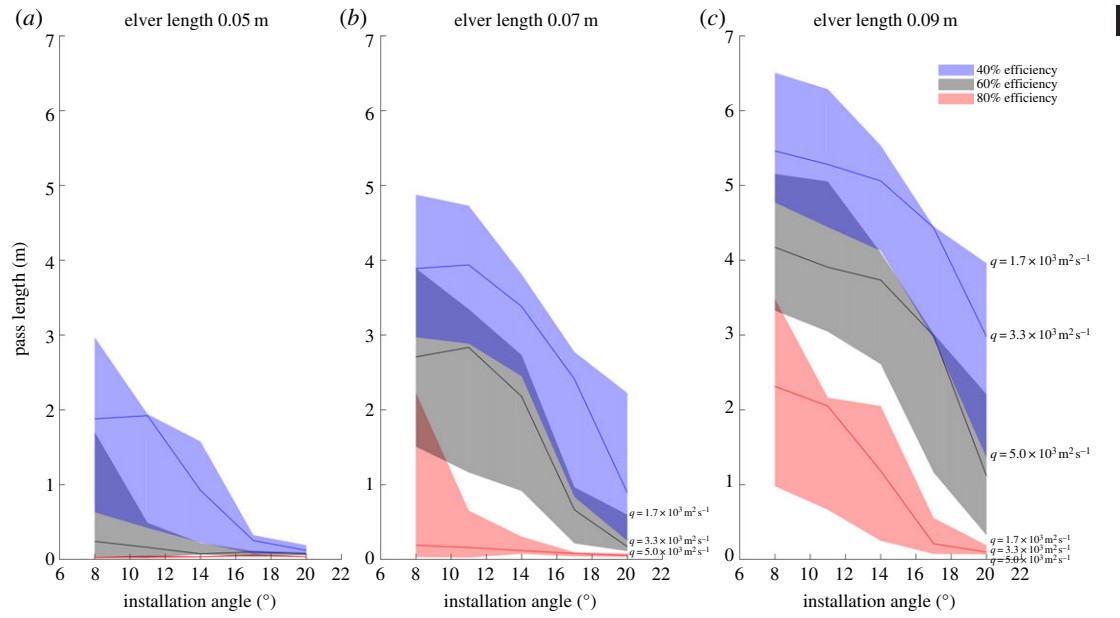

**Figure 11.** Plots of pass length against installation angle at 40%, 60% and 80% passage efficiencies of elvers of length (*a*) 0.05 m, (*b*) 0.07 m and (*c*) 0.09 m for inflow discharges per unit width of $1.67 \times 10^{-3}$, $3.33 \times 10^{-3}$ and $5.0 \times 10^{-3}$. Note that regions denoting specific passage efficiencies can overlap. For example, for 0.05 m long elvers, a 1 m long pass installed at an angle of 8° has a passage efficiency of 60% for a discharge per unit width between $1.67 \times 10^{-3}$ and $3.33 \times 10^{-3}$ m$^2$ s$^{-1}$ and a passage efficiency of 40% for a discharge per unit width between $3.33 \times 10^{-3}$ and $5.0 \times 10^{-3}$ m$^2$ s$^{-1}$. Similarly, for 0.07 m long elvers, a 2 m long pass installed at an angle of 8° has a passage efficiency of 80% for a discharge per unit width between $1.67 \times 10^{-3}$ and $3.33 \times 10^{-3}$ m$^2$ s$^{-1}$ and a passage efficiency of 60% for a discharge per unit width between $3.33 \times 10^{-3}$ and $5.0 \times 10^{-3}$ m$^2$ s$^{-1}$.

theoretical evidence that assesses the passage efficiency of eel passes and especially the impact of installation angle on passage efficiency. Furthermore, since the swimming ability of eels and elvers is related to both their body size and season of migration [32], the selected installation angle of a pass should be a function of the age of eels and elvers expected to use that pass and the distance upstream from river mouths. Although the influence of migration season has not been investigated herein, Clough *et al.* [32] presented burst swimming speeds for the spring and summer seasons: eels and elvers are significantly more energetic in spring. It is, therefore, highly unlikely that a single pass geometry is appropriate for all scenarios. Ultimately, an eel pass should be installed at the steepest angle possible while not hindering the upstream passage of the eels or elvers that are most likely to use it; there is a risk that installing eel passes at too steep an angle may cause size-selection favouring larger elvers [46].

To assist practitioners with developing improved eel pass designs, results are summarized within charts of pass length against installation angle for three lengths, or ages, of elver at selected passage efficiencies of 40%, 60% and 80% (figure 11). Although the sensitivity of passage efficiency to pass length has not been investigated in the present study, it is possible to estimate the maximum length of pass possible for each agent to ascend using the time taken to ascend the 1.25 m long pass together with the 20 s burst swimming duration used by Clough *et al.* [32] and noting that flow velocities and depths do not vary significantly along the pass. The resulting distributions were then interrogated at selected passage efficiencies. The plots can be used in five ways. First, for a given eel pass installation angle and a given passage efficiency, a practitioner can select a pass length and read off the maximum permissible discharge per unit width, $q$, that elvers of the target length (or age) can tolerate. This discharge per unit width can then be inserted into a weir equation of the form $q = \alpha(y_b + h)^{3/2}$, where $\alpha$ is a constant for the particular weir in question, $y_b$ is the elevation of the eel pass at the crest of the weir (m) and $h$ is the flow depth (m) at the eel pass entrance, to compute the necessary design elevation of the eel pass entrance. Second, for a given eel pass installation angle and a given passage efficiency, a practitioner can select a design discharge per unit width or flow depth at the eel pass entrance and read off the pass length that elvers of the target length (or age) can ascend. Third, for a given eel pass installation angle and a given pass length, a practitioner can estimate the likely passage efficiencies resulting from an imposed range of discharges per unit width or flow depths at the eel

pass entrance. Fourth, for a given eel pass length and a given passage efficiency, a practitioner can select the likely imposed range of discharges per unit width or flow depths at the eel pass entrance and read off the installation angles that elvers of the target length or age can tolerate. Finally, for a given eel pass length and a given passage efficiency, a practitioner can select a desired eel pass installation angle and read off the maximum discharge per unit width or flow depth at the eel pass entrance that elvers of the target length or age can tolerate. It is worth emphasizing that, since elvers can preferentially employ or resort to climbing when it is not possible for them to swim up a pass using anguilliform swimming alone, the passage efficiencies shown in figure 11 are conservative estimates. Conversely, it is also worth emphasizing that the estimated passage efficiencies are for spring, when eels and elvers are more energetic; passage efficiencies will be significantly lower during summer and autumn, when eels and elvers are less energetic. These sources of additional complexity and, for example, agent-to-agent interactions, may be readily incorporated within the agent-based model framework at a later date.

# 5. Conclusion

This paper reports the results of the first study to quantify the three-dimensional flow fields within fish passes composed of studded eel tiles and, furthermore, computationally assesses the upstream passage efficiency of eel tiles for juvenile European eels (*Anguilla anguilla*) using CA and individual-based models. We synthesize these unique datasets to provide, for the first time, specialists and regulatory authorities with practical criteria to inform and optimize the design and implementation of studded eel passes.

Flow fields were quantified using CFD, employing the unsteady, incompressible, RANS equations discretized with a second-order upwind scheme and using the $k$–$\omega$ SST turbulence closure model [28]. Flow depths and time- and space-averaged velocities were validated successfully against those reported by Vowles *et al.* [13]. The resulting validated model was used to compute flow fields for five installation angles (8°, 11°, 14°, 17° and 20°) and three inflow discharges per unit width ($1.67 \times 10^{-3}$, $3.33 \times 10^{-3}$ and $5.0 \times 10^{-3}$ m$^2$ s$^{-1}$). Together with reported distributions of the burst swimming speeds of eels and elvers of six different lengths (0.05, 0.06, 0.07, 0.08, 0.09 and 0.10 m), the resulting flow fields were then used as input into CA and individual- (or agent-) based models to estimate the passage efficiency of fish passes composed of studded eel tiles. Since automata were all assigned the same burst swimming speed capability, passage efficiencies estimated by the CA model tended to be near 0% or near 100% (i.e. all individuals either failed or passed) and did not accurately mimic measured values. Conversely, the individual-based model, which captured the natural variation in the swimming performance of eels and elvers, resulted in a much narrower range of passage efficiencies and accurately mimicked measured values. Overall, results suggest that passage efficiency decreases for increasing discharges and installation angles and increases for larger (and older) fish. This result is expected, since the ability to successfully pass is intrinsically linked to the water velocity within the pass, which increases with discharge and installation angle, and the speed, strength and stamina of the fish, which are greatest in spring, increase with age and should also be related to the distance upstream from the river mouth. Furthermore, results suggest that even the simplest agent-based model is able to output realistic passage efficiency values and trends and therefore agent-based modelling is well-suited to assessing the effectiveness of fish pass designs. Passage efficiencies estimated by the agent-based model can, therefore, be used with target eel and elver lengths (or ages) to improve eel pass design at specific locations through identifying optimal installation slopes and permissible inflow discharges or flow depths.

Data accessibility. The simulated flow fields associated with this paper are openly available from the University of Leeds Data Repository: https://doi.org/10.5518/710.

Authors' contributions. T.E.P. performed CFD simulations, developed the CA and individual-based model codes, analysed results and drafted the manuscript. R.E.T. analysed results, prepared figures and wrote the final manuscript. D.C.M., T.E.P. and R.E.T. conceived the project. D.J.B. and D.C.M. critically revised the manuscript. D.J.B., D.C.M. and R.E.T. supervised the project. All authors gave final approval for publication and agree to be held accountable for the work performed therein.

Competing interests. We declare we have no competing interests.

Funding. This work was supported by JBA Trust and the Engineering and Physical Sciences Research Council (grant no. EP/L01615X/1).

Acknowledgements. We wish to thank Dr Andrew Vowles of the University of Southampton, UK for sharing velocity data and video clips collected during the experiments reported by Vowles *et al.* (2015).

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
