## [Reviewer comments · Royal Society Open Science]

Review History

RSOS-191505.R0 (Original submission)

Review form: Reviewer 1 (R. Andrew Goodwin)

Is the manuscript scientifically sound in its present form?

Yes

Are the interpretations and conclusions justified by the results?

Yes

Is the language acceptable?

Yes

Do you have any ethical concerns with this paper?

No

Have you any concerns about statistical analyses in this paper?

No

Recommendation?

Accept with minor revision (please list in comments)

Comments to the Author(s)

Well written manuscript, and I applaud your work to make the results applicable to practitioners. See comments and suggestions that I provided within the attached PDF (Appendix A). More broadly, with the emphasis on "3-D" I suggest adding some figure 'zoom-ins' that showcase the patterns of 3-D flow.

Review form: Reviewer 2**Is the manuscript scientifically sound in its present form?**

Yes

Are the interpretations and conclusions justified by the results?

Yes

Is the language acceptable?

Yes

Do you have any ethical concerns with this paper?

No

Have you any concerns about statistical analyses in this paper?

No

Recommendation?

Accept with minor revision (please list in comments)

Comments to the Author(s)

Review of the manuscript RSOS-191505 submitted to Royal Society Open Science
 "Individual-based model of juvenile eel movement parameterised with Computational Fluid Dynamics-derived flow fields informs improved fish pass design"

By Thomas E. Padgett, Robert E. Thomas, Duncan J. Borman, David C. Mould,

The present work explored the flow field of fish passes composed of studded tiles by means of CFD simulation and based on the flow field, the authors studied the upstream passage efficiency of eel tiles for juvenile European eels by using cellular automata and individual-based models. The manuscript is well written and easy to follow. The findings are interesting and the work is therefore potentially suitable for publication, provided that the authors address the following issues.

The work is mainly based on the flow fields obtained by CFD approach and my major concern is related to the simulations.

Firstly, the mass conservation equation and momentum equation are not correctly given in section 2.2. The mean and fluctuating velocities can not be distinguished in equation 1 and 2. In addition, the term of gravity g_i seems not nicely defined and I would suggest to write it as $\rho g \delta y_i$, where the δ is the delta function. Please check those equations carefully. In addition, I would like to know if the inclusion of gravity would influence the flow field or not?

Secondly, the information of the mesh resolution is not given and in my view the grids should be normally refined near the boundary of studded tiles to resolve the boundary layer. I also wonder if the authors have done the test of grid-dependence. Normally the results obtained by RANS computation is sensitive to the mesh resolutions.

Lastly, I think the Reynolds numbers of different flow cases should be provided since Re is one of the most important dimensionless parameters to quantify the wall turbulent flow.

In addition, the resolution of some figures could be improved, for instance figure 4 and 8.

Decision letter (RSOS-191505.R0)

04-Nov-2019

Dear Mr Padgett

On behalf of the Editors, I am pleased to inform you that your Manuscript RSOS-191505 entitled "Individual-based model of juvenile eel movement parameterised with Computational Fluid Dynamics-derived flow fields informs improved fish pass design" has been accepted for publication in Royal Society Open Science subject to minor revision in accordance with the referee suggestions. Please find the referees' comments at the end of this email.

The reviewers and handling editors have recommended publication, but also suggest some minor revisions to your manuscript. Therefore, I invite you to respond to the comments and revise your manuscript.

- Ethics statement

- Data accessibility

<http://datadryad.org/submit?journalID=RSOS&manu=RSOS-191505>

- Competing interests

- Authors' contributions

- Acknowledgements

- Funding statement

Because the schedule for publication is very tight, it is a condition of publication that you submit the revised version of your manuscript before 13-Nov-2019. Please note that the revision deadline will expire at 00.00am on this date. If you do not think you will be able to meet this date please let me know immediately.

Kind regards,
Andrew Dunn
Senior Publishing Editor
Royal Society Open Science
openscience@royalsociety.org

on behalf of Dr Francois Fages (Associate Editor) and Kevin Padian (Subject Editor)
openscience@royalsociety.org

Associate Editor Comments to Author (Dr Francois Fages):

Associate Editor: 1

Comments to the Author:

Dear authors

It is my pleasure to accept your paper with minor revision. Nevertheless it is important that you do take into account the comments, corrections and annotations of both reviewers in your final submission.

Best regards

Reviewer comments to Author:

Reviewer: 1

Comments to the Author(s)

Well written manuscript, and I applaud your work to make the results applicable to practitioners. See comments and suggestions that I provided within the attached PDF. More broadly, with the emphasis on "3-D" I suggest adding some figure 'zoom-ins' that showcase the patterns of 3-D flow.

Reviewer: 2

Comments to the Author(s)

Review of the manuscript RSOS-191505 submitted to Royal Society Open Science

"Individual-based model of juvenile eel movement parameterised with Computational Fluid Dynamics-derived flow fields informs improved fish pass design"

By Thomas E. Padgett, Robert E. Thomas, Duncan J. Borman, David C. Mould,

The present work explored the flow field of fish passes composed of studded tiles by means of CFD simulation and based on the flow field, the authors studied the upstream passage efficiency of eel tiles for juvenile European eels by using cellular automata and individual-based models. The manuscript is well written and easy to follow. The findings are interesting and the work is therefore potentially suitable for publication, provided that the authors address the following issues.

The work is mainly based on the flow fields obtained by CFD approach and my major concern is related to the simulations.

Firstly, the mass conservation equation and momentum equation are not correctly given in section 2.2. The mean and fluctuating velocities can not be distinguished in equation 1 and 2. In addition, the term of gravity g_i seems not nicely defined and I would suggest to write it as $\rho g \delta_{y,i}$, where the δ is the delta function. Please check those equations carefully. In addition, I would like to know if the inclusion of gravity would influence the flow field or not? Secondly, the information of the mesh resolution is not given and in my view the grids should be normally refined near the boundary of studded tiles to resolve the boundary layer. I also wonder if the authors have done the test of grid-dependence. Normally the results obtained by RANS computation is sensitive to the mesh resolutions.

Lastly, I think the Reynolds numbers of different flow cases should be provided since Re is one of the most important dimensionless parameters to quantify the wall turbulent flow.

In addition, the resolution of some figures could be improved, for instance figure 4 and 8.

Author's Response to Decision Letter for (RSOS-191505.R0)

See Appendix B.

Decision letter (RSOS-191505.R1)

27-Nov-2019

Dear Mr Padgett,

It is a pleasure to accept your manuscript entitled "Individual-based model of juvenile eel movement parameterised with Computational Fluid Dynamics-derived flow fields informs improved fish pass design" in its current form for publication in Royal Society Open Science. The comments of the reviewer(s) who reviewed your manuscript are included at the foot of this letter.

on behalf of Dr Francois Fages (Associate Editor) and Kevin Padian (Subject Editor)
openscience@royalsociety.org

Appendix A**ROYAL SOCIETY
OPEN SCIENCE****Individual-based model of juvenile eel movement
parameterised with Computational Fluid Dynamics-derived
flow fields informs improved fish pass design**

Journal:	Royal Society Open Science
Manuscript ID	RSOS-191505
Article Type:	Research
Date Submitted by the Author:	17-Sep-2019
Complete List of Authors:	Padgett, Thomas; Centre for Doctoral Training in Fluid Dynamics, University of Leeds Thomas, Robert; Energy and Environment Institute, University of Hull Borman, Duncan; School of Civil Engineering, University of Leeds Mould, David; JBA Consulting
Subject:	Computer modelling and simulation < COMPUTER SCIENCE, ecology < BIOLOGY
Keywords:	Cellular Automata, Computational Fluid Dynamics, Eel Tiles, European Eel, Individual-based Modelling
Subject Category:	Biology (whole organism)

Author-supplied statements

Relevant information will appear here if provided.

Ethics

Does your article include research that required ethical approval or permits?:

This article does not present research with ethical considerations

Statement (if applicable):

CUST_IF_YES_ETHICS :No data available.

Data

It is a condition of publication that data, code and materials supporting your paper are made publicly available. Does your paper present new data?:

Yes

Statement (if applicable):

The simulated flow fields and models associated with this paper are openly available from the University of Leeds Data Repository: <https://doi.org/10.5518/710>

Conflict of interest

I/We declare we have no competing interests

Statement (if applicable):

CUST_STATE_CONFLICT :No data available.

Authors' contributions

This paper has multiple authors and our individual contributions were as below

Statement (if applicable):

T.E.P. performed CFD simulations, developed the CA and individual-based model codes, analysed results and drafted the manuscript. R.E.T. analysed results, prepared figures and wrote the final manuscript. D.C.M., T.E.P. and R.E.T. conceived the project. D.J.B. and D.C.M. critically revised the manuscript. D.J.B., D.C.M. and R.E.T. supervised the project. All authors gave final approval for publication and agree to be held accountable for the work performed therein.

Individual-based model of juvenile eel movement parameterised with Computational Fluid Dynamics-derived flow fields informs improved fish pass design

Thomas E. Padgett^{1*}, Robert E. Thomas², Duncan J. Borman³, David C. Mould⁴,

^{1*} Corresponding author. Centre for Doctoral Training in Fluid Dynamics, University of Leeds, LS2 9JT, UK. email: ed10tep@leeds.ac.
[revised manuscript text omitted]

$$\frac{\partial}{\partial t}(\rho \bar{u}_i) + \frac{\partial}{\partial x_j}(\rho \bar{u}_i \bar{u}_j) - g_i + \frac{\partial P}{\partial x_i} - \mu \frac{\partial^2 \bar{u}_i}{\partial x_j^2} - \frac{\partial}{\partial x_j}(-\rho \overline{u_i' u_j'}) = 0 \quad (2)$$

where equation (1) is the mass conservation equation and equation (2) is the momentum conservation equations and \mathbf{u} = velocity (m s^{-1}), \mathbf{x} = displacement (m), t = time (s), ρ = density of water ($\sim 997.8 \text{ kg m}^{-3}$ at 21.8°C), \mathbf{g} = gravitational acceleration vector [$-g \sin \theta$, $-g \cos \theta$, 0], P = pressure (Pa), μ = dynamic viscosity of water ($\sim 9.58 \times 10^{-4} \text{ Pa}\cdot\text{s}$ at 21.8°C), the indices i and j ($i, j = 1$ to 3) denote the three components of displacement (x, y, z) and velocity (u, v, w), overbars denote time averages and primes denote fluctuations about those averages. The momentum conservation equations require a turbulence closure for the Reynolds stress term $\frac{\partial}{\partial x_j}(-\rho \overline{u_i' u_j'})$. The k - ω SST model was used [28], which combines the robust and accurate formulation of the k - ω model in the near-wall region with the freestream independence of the k - ϵ model in the far field [29]. The k - ω SST model exhibits enhanced performance relative to conventional k - ϵ and k - ω models when applied to adverse pressure gradient flows, airfoils, and sub- to super-critical transitions [29]. The temporal gradients and the advection terms in equations (2), the turbulent kinetic energy, k , and specific turbulence dissipation rate, ω , were discretised using a second order upwind scheme. The SIMPLE scheme [30] was used to couple the velocities and the pressure. The free surface location was approximated using the Volume of Fluid (VOF) method [31] with two Eulerian phases, water and air, and using the implicit body force formulation. The free surface was interpolated using Fluent's Geo-Reconstruct scheme, which fits a piecewise-linear interface within each cell and uses that linear shape to estimate the advection of fluid through the cell faces in a three-step procedure [27].

Water entered the domain through a velocity inlet positioned upstream of the eel pass and left the domain through a pressure outlet located downstream of the pass (figure 2). All other boundaries were defined as walls with a no-slip boundary condition and a roughness of 0. A gravitational acceleration term, with components in the x and y directions to define the installation angle, was applied to the domain. Simulations were undertaken with a constant time step, Δt , of 5.0×10^{-4} seconds. The convergence criterion for the non-dimensional residuals of u , v , and w , mass continuity, turbulent kinetic energy, k , and specific turbulence dissipation rate, ω was defined as 10^{-4} . A maximum limit of 100 iterations per time step was imposed.

2.2 Quantifying eel tile "passability"

In order to assess the "passability" of eel tiles at different flow rates and inclinations, it was necessary to compare computed 3-D flow fields against the swimming and/or climbing abilities of eelers. In the absence of data detailing climbing performance, the burst swimming performance data of Clough et al. [32] were used. These data were obtained from 417 eelers collected from the River Severn in April (mean length $67.61 \pm 0.57 \text{ mm}$; mean water temperature $11.1 \pm 0.32^\circ\text{C}$) and

1
2
3 in June/July (mean length 147.76 ± 4.81 mm; mean water temperature $18.64 \pm 0.12^\circ\text{C}$) [32].

[revised manuscript text omitted]

398.
- Tesch, F. (2004). *The Eel*, 5th ed., Blackwell Publishing, Oxford, UK.
- Poole, W.R., Reynolds, J.D. (1998). Variability in growth rate in European eel *Anguilla anguilla*
(L.) in a western Irish catchment. *P. Roy. Irish Acad. B* 98(3), 141-145.
- Moriarty, C. (1986). Variations in elver abundance at European catching stations from 1938
to 1985 (*Anguilla anguilla*). *Vie Milieu* 36, 233-235.
- Moriarty, C. (1996). The decline in catches of European elver 1980-1992. *Archiwum*
*Rybactwa Polskiego* 4(2a), 245-248.
- Jacoby, D., Gollock, M. (2014). *Anguilla anguilla*. The IUCN Red List of Threatened Species
2014: e.T60344A45833138. <https://www.iucnredlist.org/species/60344/45833138>
(accessed 05/07/19).

Moriarty, C., Dekker, W. (eds.) (1997). Management of the European eel. Second report of EC
Concerted Action AIR A94-1939: Enhancement of the European eel fishery and conservation
of the species. Irish Marine Institute Fisheries Bulletin 15. Marine Institute, Dublin.
Feunteun, E. (2002). Management and restoration of European eel population (*Anguilla*
*anguilla*): An impossible bargain. *Ecol. Eng.* 18(5), 575-591.
Solomon, D., Beach, M. (2004). Fish pass design for Eel and Elver (*Anguilla anguilla*). R&D
Technical Report W2-070TR. Environment Agency RND report. UK Government Publishing
House, London.
Knights, B., White, E.M. (1998). Enhancing immigration and recruitment of eels: the use of
passes and associated trapping systems. *Fisheries Manag. Ecol.* 5(6), 459-471.
Briand, C., Fatin, D., Fontenelle, G., Feunteun, E. (2005). Effect of re-opening of a migratory
pathway for eel (*Anguilla anguilla*, L.) at a watershed scale. *B. Fr. Pêche Piscic.* 378-379, 67-
86.
Piper, A.T., Wright, R.M., Kemp, P.S. (2012). The influence of attraction flow on upstream
passage of European eel (*Anguilla anguilla*) at intertidal barriers. *Ecol. Eng.* 44, 329-336.
Vowles, A.S., Don, A., Karageorgopoulos, P., Worthington, T., Kemp, P. (2015). Efficiency of a
dual density studded fish pass designed to mitigate for impeded upstream passage of juvenile
European eels (*Anguilla anguilla*) at a model crump weir. *Fisheries Manag. Ecol.* 22(4), 307-
316.
Porcher, J. (2002). Fishways for eels. *B. Fr. Pêche Piscic.* 364, 147-155.
Jackman, G., Larson, M., Ruzicka, V. (2009). American eel passage enhancement plan for the
Bronx River. Report for Park & Recreation department, City of New York, NY.
Environment Agency (2011). Elver and eel passes: A guide to the design and implementation
of passage solutions at weirs, tidal gates and sluices. Environment Agency report number
GEH00211BTMV-E-E. Environment Agency, Bristol.
Voegtle, B., Larinier, M. (2000). Etude sur les capacités de franchissement des civelles et
anguillettes: Site hydroélectrique de Tuilières sur la Dordogne (24) barrage estuarien d'Arzal
sur la Vilaine (56). CEMAGREF GHAAPE Rep. RA00.05. CEMAGREF, Toulouse.
Berry & Escott Engineering (2017). Berry & Escott Engineering: Eel Tile Product Page.
<http://www.berrypescott.co.uk/eel-tile/> (Accessed 07/02/17).
Milieu Inc. (2017). Milieu Products: Substrates for Elvers.
<http://www.milieuinc.com/products> (Accessed 07/02/17).
Terraqua Environmental Solutions (2017). Terraqua: Eel Substrates. <http://terraqua->
[es.co.uk/fish-pass/eel-substrates](http://terraqua-es.co.uk/fish-pass/eel-substrates) (Accessed 07/02/17).
Bombač, M., Novak, G., Rodič, P., Četina, M. (2014). Numerical and physical model study of a
vertical slot fishway. *J. Hydrol. Hydromech.* 62(2), 150-159.

Gisen, D.C., Weichert, R.B., Nestler, J.M. (2016). Optimizing attraction flow for upstream fish
passage at a hydropower dam employing 3-D detached-eddy simulation. *Ecol. Eng.* 100, 344-
353.
Marriner, B.A., Baki, A.B., Zhu, D.Z., Cooke, S.J., Katopodis, C. (2016). The hydraulics of a
vertical slot fishway: A case study on the multi-species Vianney-Legendre fishway in Quebec,
Canada. *Ecol. Eng.* 90, 190-202.
Chen, S.C., Wang, S.C., Tfwala, S.S. (2017). Hydraulics driven upstream migration of
Taiwanese indigenous fishes in a fish-bone-type fishway. *Ecol. Eng.* 108A, 179-193. doi:
10.1016/j.ecoleng.2017.08.015
Goodwin, R.A., Politano, M., Garvin, J.W., Nestler, J.M., Hay, D., Anderson, J.J., Weber, L.J.,
Dimperio, E., Smith, D.L., Timko, M. (2014). Fish navigation of large dams emerges from their
modulation of flow field experience. *P. Natl. Acad. Sci.* 111(14), 5277-5282.
Gao, Z., Andersson, H.I., Dai, H., Jiang, F., Zhao, L. (2016). A new Eulerian-Lagrangian Agent
Method to model fish paths in a vertical slot fishway. *Ecol. Eng.* 88, 217-225.
ANSYS, Inc. (2016). ANSYS Fluent Theory Guide, Release 17.2, August 2016. ANSYS, Inc.,
Canonsburg, PA.
Menter, F.R., Kuntz, M., Langtry, R. (2003). Ten years of industrial experience with the SST
turbulence model, in: Hanjalić, K., Nagano, Y., Tummers, M. (Eds.), *Turbulence, Heat and Mass*
*Transfer 4, Proceedings of the Fourth International Symposium on Turbulence, Heat and*
*Mass Transfer, Antalya, Turkey, 12-17 October, 2003.* Begell House, Inc., New York, NY, pp.
625-632.
Menter, F.R. (2009). Review of the shear-stress transport turbulence model experience from
an industrial perspective. *Int. J. Comp. Fluid D.* 23(4), 305-316.
Patankar, S.V., Spalding, D.B. (1972). A calculation procedure for heat, mass and momentum
transfer in three-dimensional parabolic flows. *Int. J. Heat Mass Tran.* 15(10), 1787-1806.
Hirt, C., Nichols, B. (1981). Volume of fluid (VOF) method for the dynamics of free boundaries.
*J. Comput. Phys.* 39(1), 201-225.
Clough, S.C., Lee-Elliott, I.H., Turnpenny, A.W.H., Holden, S.D.J., Hinks, C. (2004). *Swimming*
*Speeds in Fish: Phase 2.* Environment Agency R&D Technical Report W2-049/TR1.
Environment Agency, Bristol.
Deelder, C.L. (1984). Synopsis of biological data on the eel, *Anguilla anguilla* (Linnaeus,
1758). Food and Agriculture Organization of the United Nations Fisheries Synopsis 80,
Revision 1. Food and Agriculture Organization of the United Nations, Rome.
Moore, E.F. (1962). Machine models of self-reproduction. *Proc. Sym. Ap.* 14, 17-33.

Matsumoto, M., Nishimura, T. (1998). Mersenne twister: a 623-dimensionally
equidistributed uniform pseudo-random number generator. *ACM T. Model. Comput. S.* 8(1),
3-30.
Vowles, A.S., Don, A.M., Karageorgopoulos, P., Kemp, P.S. (2017). Passage of European eel and
river lamprey at a model weir provisioned with studded tiles. *Journal of Ecohydraulics* 2(2),
88-98. doi: 10.1080/24705357.2017.1310001
Moulinec, C., Hunt, J.C.R., Nieuwstadt, F.T.M. (2004). Disappearing wakes and dispersion in
numerically simulated flows through tube bundles. *Flow. Turbul. Combust.* 73(2), 95-116.
Tong, F., Cheng, L., Zhao, M., Zhou, T., Chen, X. (2014). The vortex shedding around four
circular cylinders in an in-line square configuration. *Phys. Fluids* 26, 024112. doi:
10.1063/1.4866593.
Benhamadouche, S., Laurence, D. (2003). LES, coarse LES, and transient RANS comparisons
on the flow across a tube bundle. *Int. J. Heat Fluid Fl.* 24(4), 470-479.
Launder, B.E., Spalding, D.B. (1974). The numerical computation of turbulent flows. *Comput.*
*Method. Appl. M.* 3(2), 269-289.
Ingham, D.B., Ma, L. (2005). Fundamental equations for CFD in river flow simulations, in:
Bates, P.D., Lane, S.N., Ferguson, R.I. (Eds.), *Computational Fluid Dynamics: Applications in*
*Environmental Hydraulics*. John Wiley & Sons, Chichester, UK, pp. 19-49.
Wilcox, D.C. (1988). Reassessment of the scale-determining equation for advanced
turbulence models. *AIAA J.* 26(11), 1299-1310.
Piper, A.T., Manes, C., Siniscalchi, F., Marion, A., Wright, R.M., Kemp, P.S. (2015). Response of
seaward-migrating European eel (*Anguilla anguilla*) to manipulated flow fields. *P. Roy. Soc B*
282, 20151098. doi: 10.1098/rspb.2015.1098
Jellyman, D.J. (1977). Invasion of a New Zealand freshwater stream by glass-eels of two
*Anguilla* spp. *New Zeal. J. Mar. Fresh.* 11(2), 193-209. doi: 10.1080/00288330.1977.9515673
Legault, A. (1988). Le franchissement des barrages par l'escalade de l'anguille; étude en Sèvre
Niortais. *B. Fr. Pêche Piscic.* 308, 1-10.
Jellyman, P.G., Bauld, J.T., Crow, S.K. (2017). The effect of ramp slope and surface type on the
climbing success of shortfin eel (*Anguilla australis*) elvers. *Mar. Freshwater Res.* 68(7), 1317-
1324. doi: 10.1071/MF16015

Funding

This work was supported by JBA Trust and the Engineering and Physical Sciences
Research Council [grant number EP/L01615X/1].

Data, code and materials

The models and simulated flow fields associated with this paper are openly available from the University of Leeds Data Repository: <https://doi.org/10.5518/710>.

Competing interests

We declare we have no competing interests.

Authors' contributions

T.E.P. performed CFD simulations, developed the CA and individual-based model codes, analysed results and drafted the manuscript. R.E.T. analysed results, prepared figures and wrote the final manuscript. D.C.M., T.E.P. and R.E.T. conceived the project. D.J.B. and D.C.M. critically revised the manuscript. D.J.B., D.C.M. and R.E.T. supervised the project. All authors gave final approval for publication and agree to be held accountable for the work performed therein.

List of Figures

Figure 1: Typical dual density studded eel tile [19] (Photograph by T. E. Padgett, 14/06/17).

Figure 2: Isometric view of the CFD domain. Cyan denotes no-slip boundary condition; magenta denotes symmetry boundary condition; black denotes pressure outlet boundary condition.

Figure 3: Median, 10th and 90th percentile burst swimming speed in spring, at a water temperature of 21.8 °C against eel and elver body length from the SWIMIT 3.3 model [32]. Solid line depicts the median burst swimming speed while dotted lines show the 10th and 90th percentile burst swimming speeds.

Figure 4: Classified time-averaged velocity fields extracted in a plane 3 mm above the floor of an eel tile installed at 11° at a discharge per unit width of $3.33 \times 10^{-3} \text{ m}^2\text{s}^{-1}$. Classified using the 10th, 50th and 90th percentile burst swimming speeds for elver of length 0.05, 0.06, 0.07, 0.08, 0.09, and 0.10 m, respectively. Elver length (in m) is shown to the left. White denotes “passable”, red denotes “impassable”, black denotes “studs”, and grey denotes “boundary”.

Figure 5: Sensitivity of passage efficiency of cellular automata (CA) agents to: A) number of
spawned CA agents and B) number of timesteps for elver of length 0.07 m ascending an eel
pass installed on a 1.25 m-long model crump weir inclined at 11° and with a unit discharge of
$3.33 \times 10^{-3} \text{ m}^2\text{s}^{-1}$. All simulations in A were performed over 10000 timesteps, while all
simulations in B were performed using 1000 CA agents.

Figure 6: Example automaton movement. Blue circles represent automata, white denotes "passable", red denotes "impassable" and grey denotes "boundary" cells.

Figure 7: Sensitivity of passage efficiency of cellular automata (CA) agents to: A) number of
timesteps before a CA agent is classified as “stuck” and B) number of timesteps a CA agent that
is classified as “stuck” before it is forced to “fall back” before recommencing ascent. Both
simulations performed for elver of length 0.07 m ascending an eel pass installed on a 1.25 m-
long model crump weir inclined at 17° and with a unit discharge of $5.0 \times 10^{-3} \text{ m}^2\text{s}^{-1}$, using 1000
agents and 10000 timesteps; simulations in B performed with the number of timesteps before
an agent is classified as “stuck” set equal to 20.

Figure 8: A) Isometric and B) plan views of the pass with an installation angle of 11° and a discharge per unit width of $3.33 \times 10^{-3} \text{ m}^2\text{s}^{-1}$, with overlain free surface after 10.0 seconds of flow time. The free surface is overlain with contours of velocity magnitude at the free surface. The domain has been cropped to increase ease of viewing.

Figure 10: Passage efficiency (in %) predicted by the CA model assuming 10th, 50th and 90th percentile burst swimming speeds and the individual-based model against elver length for eel tiles installed at a range of installation angles at three different discharges per unit width.

Figure 11: Plots of pass length against installation angle at 40%, 60% and 80% passage efficiencies of elvers of length A) 0.05 m, B) 0.07 m and C) 0.09 m for inflow discharges per unit width of $1.67 \times 10^{-3} \text{ m}^2\text{s}^{-1}$, $3.33 \times 10^{-3} \text{ m}^2\text{s}^{-1}$ and $5.0 \times 10^{-3} \text{ m}^2\text{s}^{-1}$. Note that regions denoting specific passage efficiencies can overlap. For example, for 0.05 m-long elvers, a 1 m-long pass installed at an angle of 8° has a passage efficiency of 60% for a discharge per unit width between $1.67 \times 10^{-3} \text{ m}^2\text{s}^{-1}$ and $3.33 \times 10^{-3} \text{ m}^2\text{s}^{-1}$ and a passage efficiency of 40% for a discharge per unit width between $3.33 \times 10^{-3} \text{ m}^2\text{s}^{-1}$ and $5.0 \times 10^{-3} \text{ m}^2\text{s}^{-1}$. Similarly, for 0.07 m-long elvers, a 2 m-long pass installed at an angle of 8° has a passage efficiency of 80% for a discharge per unit width between $1.67 \times 10^{-3} \text{ m}^2\text{s}^{-1}$ and $3.33 \times 10^{-3} \text{ m}^2\text{s}^{-1}$ and a passage efficiency of 60% for a discharge per unit width between $3.33 \times 10^{-3} \text{ m}^2\text{s}^{-1}$ and $5.0 \times 10^{-3} \text{ m}^2\text{s}^{-1}$.

Tables

Table 1: Passage efficiency estimated using the cellular automata and individual-based models of eelers of length 0.07 m ascending a 1.25 m-long eel pass inclined at 11° and with an inflow discharge per unit width of $3.33 \times 10^{-3} \text{ m}^2\text{s}^{-1}$, compared to the passage efficiencies reported by Vowles et al. (2015).

Model	Percentile eel burst swimming speed		
	10%	50%	90%
Cellular automata	0.0%	97.8%	97.8%
Individual-based	75.5%		
Vowles et al. (2015)	Total pass		66.7%
	Small studs only		73.8%

Appendix B

Thomas Padgett
Doctoral Research Student
EPSRC CDT in Fluid Dynamics
School of Computing,
University of Leeds,
Leeds, LS2 9JT.
UK
Email: ed10tep@leeds.ac.uk, thomas.e.padgett@outlook.com

UNIVERSITY OF LEEDS

22nd November 2019

RE: Resubmission of manuscript RSOS-191505

We are pleased to submit a revised version of the manuscript entitled: “Individual-based model of juvenile eel movement parameterised with Computational Fluid Dynamics-derived flow fields informs improved fish pass design” by Thomas E. Padgett, Robert E. Thomas, Duncan J. Borman and David C. Mould. We have addressed the points raised the reviewers. As requested, we have uploaded a “tracked changes” version of the manuscript (filename: Padgett et al_Marked_Up.docx) and a clean “final” version of the manuscript (filename: Padgett et al_Revised_Final.docx).

Below you will find a list of the reviewers’ observations and the corresponding changes made, marked in red. We trust that we have adequately addressed the editing comments and that the revised manuscript is improved as a result.

Regards,

Thomas Padgett

Decision Letter (RSOS-191505)

04-Nov-2019

Dear Mr Padgett,

On behalf of the Editors, I am pleased to inform you that your Manuscript RSOS-191505 entitled "Individual-based model of juvenile eel movement parameterised with Computational Fluid Dynamics-derived flow fields informs improved fish pass design" has been accepted for publication in Royal Society Open Science subject to minor revision in accordance with the referee suggestions. Please find the referees' comments at the end of this email.

The reviewers and handling editors have recommended publication, but also suggest some minor revisions to your manuscript. Therefore, I invite you to respond to the comments and revise your manuscript.

Associate Editor Comments to Author (Dr Francois Fages):

Associate Editor: 1

Comments to the Author:

Dear authors

It is my pleasure to accept your paper with minor revision. Nevertheless it is important that you do take into account the comments, corrections and annotations of both reviewers in your final submission.

Best regards

We would like to thank the AE for their supportive comments. In addition to the corrections suggested by the reviewers, we noticed that references 39-42 had been transposed during the conversion from Harvard referencing to Vancouver referencing; we have corrected this. In addition, in order to reduce the manuscript to the 8,000 word limit of Interface (to where we initially submitted the manuscript), we deleted a sentence at the end of the first paragraph of the conclusion: "We synthesise these unique datasets to provide, for the first time, specialists and regulatory authorities with practical criteria to inform and optimise the design and implementation of studded eel passes". We feel that this sentence improves the linkage between the results and discussion and would therefore like to reintroduce it at the discretion of the AE. Many thanks for your consideration.

Reviewer comments to Author:

Reviewer: 1

Comments to the Author(s)

Well written manuscript, and I applaud your work to make the results applicable to practitioners. See comments and suggestions that I provided within the attached PDF. More broadly, with the emphasis on "3-D" I suggest adding some figure 'zoom-ins' that showcase the patterns of 3-D flow. We thank the reviewer for their positive and supportive comments.

Lines 272-273: we have attempted to clarify the statement: "Any deviation from these trends is thought to be due to instability within the CFD solutions at a discharge per unit width of $1.67 \times 10^{-3} \text{ m}^2\text{s}^{-1}$ " by modifying it to: "Any deviation from these trends is thought to be due to numerical instabilities within the CFD solutions caused by the combination of very small flow depths and fast velocities at a discharge per unit width of $1.67 \times 10^{-3} \text{ m}^2\text{s}^{-1}$ ".

In response to the suggestion to add "zoom ins" to figures 2 and 8, we have added insets to both figures showing magnified regions of the computational mesh and of the water surface, respectively.

We have modified the caption to Figure 4 to emphasise that boundary cells are one cell thick and thus appear as lines: “Note that boundaries are one cell thick and so appear as lines in the figure”.

We thank the reviewer for their praise of Figure 6 and suggestion to move it earlier in the manuscript. However, we have re-read the manuscript repeatedly and cannot find a suitable location without disrupting the flow of the text.

We have redrawn the lines in figure 10 in an attempt to improve contrast between the lines.

Reviewer: 2

Comments to the Author(s)

Review of the manuscript RSOS-191505 submitted to Royal Society Open Science
“Individual-based model of juvenile eel movement parameterised with Computational Fluid Dynamics-derived flow fields informs improved fish pass design”

By Thomas E. Padgett, Robert E. Thomas, Duncan J. Borman, David C. Mould,

The present work explored the flow field of fish passes composed of studded tiles by means of CFD simulation and based on the flow field, the authors studied the upstream passage efficiency of eel tiles for juvenile European eels by using cellular automata and individual-based models. The manuscript is well written and easy to follow. The findings are interesting and the work is therefore potentially suitable for publication, provided that the authors address the following issues.

We thank the reviewer for their positive comments.

The work is mainly based on the flow fields obtained by CFD approach and my major concern is related to the simulations.

Firstly, the mass conservation equation and momentum equation are not correctly given in section 2.2. The mean and fluctuating velocities can not be distinguished in equation 1 and 2.

Our apologies: upon upload and conversion to pdf, equations 1 and 2 corrupted and the overbars that were present in our Word document disappeared!

In addition, the term of gravity g_i seems not nicely defined and I would suggest to write it as $\rho g \delta_{y,i}$, where the δ is the delta function. Please check those equations carefully.

We thank the reviewer for noting the missing ρ in equation 2 and have corrected this. In addition, we have elected to rewrite the equations in non-conservation form since we are employing the incompressible Navier-Stokes equations. This allows us to divide all terms in equation 2 by ρ since ρ does not vary in space nor time. However, the suggestion to employ the delta function is erroneous, since the gravitational term is decomposed into its slope-parallel and slope-perpendicular components.

In addition, I would like to know if the inclusion of gravity would influence the flow field or not?

Gravity is explicitly incorporated within our simulations through decomposing the gravitational acceleration vector into its slope-parallel and slope-perpendicular components. We believe that the reviewer’s query above and this query stem from the order FLUENT lists its coordinate system: x = slope-parallel, y = slope-perpendicular and z = spanwise, which contrasts with typical usage in environmental science where the coordinate system is generally listed as: x = slope-parallel, y = spanwise and z = slope-perpendicular. To address this potential confusion, we have emphasised the definition of the coordinate system in lines 116-117 and also emphasised the benefit of defining the gravitational acceleration vector as $[-g \sin \theta, -g \cos \theta, 0]$ (i.e., slope-parallel, slope-perpendicular, spanwise components) in lines 136-138.

Secondly, the information of the mesh resolution is not given and in my view the grids should be normally refined near the boundary of studded tiles to resolve the boundary layer.

In response to reviewer 1, we have added an inset to figure 2 showing a magnified version of the mesh. We have also added details of the mesh composition and resolution to lines 101-106: “The geometry required use of an unstructured tetrahedral mesh featuring approximately 906000 cells, with increased cell density at the studs and at the bed (figure 2b). To adequately capture the boundary layer, an “inflation layer” [27] was applied to the bed and to the walls of each stud. This split the near-wall region into five sub-layers, each of which was 1.2 times larger than the previous sub-layer. Thus, sub-layer thickness transitioned smoothly from a near-wall cell thickness of ~ 0.4 mm to a thickness of 1.0 mm at a distance of 3.0 mm from the wall.”

I also wonder if the authors have done the test of grid-dependence. Normally the results obtained by RANS computation is sensitive to the mesh resolutions.

Grid dependence tests were undertaken using meshes featuring 650000 cells and 1300000 cells. Within the lowermost 3 mm of the domain used in our paper, these tests suggested a mean error of 2.5% on the velocity magnitude for the mesh featuring 906000 cells. While this is perfectly acceptable, we feel that reporting this within the paper would deviate and distract from the narrative and so we favour not reporting it.

Lastly, I think the Reynolds numbers of different flow cases should be provided since Re is one of the most important dimensionless parameters to quantify the wall turbulent flow.

We have added Reynolds numbers to section 3.1 (lines 228-233): “These simulated values yield Reynolds numbers of 1740 or 4550 using the mean flow depth or the stud diameter as the length scale, respectively. In other words, the flow is fully turbulent [37-38]. Indeed, flows are fully turbulent for all simulated cases, with Reynolds numbers ranging from 660 or 2320 for the 8° , $1.67 \times 10^{-3} \text{ m}^2\text{s}^{-1}$ case to 3070 or 5800 for the 20° , $5.0 \times 10^{-3} \text{ m}^2\text{s}^{-1}$ case using the mean flow depth or stud diameter as the length scale, respectively.”

In addition, the resolution of some figures could be improved, for instance figure 4 and 8.

Figure quality was degraded when our figures were pasted into Word for single-file upload. We have uploaded vector graphics or high-resolution versions of all figures as part of this revision. In particular, we have modified figure 6 so that it can be printed in grayscale and recoloured figure 8 so that it is colour-safe.